# Influenza B Virus (IBV) Immune-Mediated Disease in C57BL/6 Mice

**DOI:** 10.3390/vaccines10091440

**Published:** 2022-09-01

**Authors:** Harrison C. Bergeron, Zachary Beau Reneer, Aakash Arora, Stephen Reynolds, Tamas Nagy, Ralph A. Tripp

**Affiliations:** 1Department of Infectious Diseases, College of Veterinary Medicine, University of Georgia, Athens, GA 30605, USA; 2Department of Pathology, College of Veterinary Medicine, University of Georgia, Athens, GA 30605, USA

**Keywords:** influenza B virus, IBV, host–pathogen interaction, immune-mediated disease, dexamethasone, mouse model

## Abstract

Influenza B viruses (IBV) primarily infect humans, causing seasonal epidemics. The absence of an animal reservoir limits pandemic concern, but IBV infections may cause severe respiratory disease, predominantly in young children and the elderly. The IBV disease burden is largely controlled by seasonal influenza vaccination; however, immunity due to vaccination is sometimes incomplete, a feature linked to antigenic mismatches. Thus, understanding the features that contribute to disease pathogenesis is important, particularly immune-mediated versus virus-mediated outcomes. Unexpectedly, C57BL/6 (B6) mice intranasally infected with a low multiplicity of infection of B/Florida/04/2006 developed substantial morbidity and mortality. To address the cause, B6 mice were treated daily with dexamethasone to dampen the immune and pro-inflammatory response to IBV infection, allowing the determination of whether the responses were immune- and/or virus-associated. As expected, dexamethasone (DEX)-treated mice had a lower pro-inflammatory response and reduced lung pathology despite the presence of high viral lung titers, but mortality was comparable to PBS-treated mice, indicating that mortality may be linked to lung virus replication. The results showed that the immune response to IBV is the major cause of morbidity, mortality, lung pathology, and viral clearance. Importantly, the results suggest that a robust lung CTL response and associated leukocyte influx contribute to disease.

## 1. Introduction

Influenza A and B viruses (IAV and IBV) circulate annually, causing worldwide seasonal epidemics. IAV and IBV may cause epidemics or pandemics that may result in several millions of respiratory disease cases and approximately 250,000–650,000 deaths annually; IBV infection averages a quarter of the annual influenza disease burden [1] but has occasionally caused up to ~80% of infections [2]. IBV is frequently perceived as a less severe influenza infection, but this is likely because the epidemiology and clinical outcomes of IBV have been less thoroughly investigated in hospitalized patients [3,4]. IBV is relatively understudied compared to IAV, and it is thought to be less prevalent and infect mostly younger hospitalized patients compared to IAV [5]; however, both strains contribute to the hospitalization burden and complications.

IAV and IBV are considerably related but molecularly distinct in their viral proteins, their tropisms, and the antiviral responses they induce [6,7]. IBVs have split into two lineages, i.e., B/Yamagata and B/Victoria, and like IAVs, IBVs are divided into clades and sub-clades. Replication of IAV and IBV is similar—they both infect and replicate in respiratory epithelial cells, and both viruses bind to sialic acid receptors on host cells by their hemagglutinin (HA) attachment proteins. Binding results in virus–cell fusion and entry into the cell’s endosomes, resulting in the release of eight viral ribonucleoprotein (vRNP) complexes needed for nuclear replication [8]. However, IBV lacks the expression of the PB1-F2 and PA-X virulence factors found in IAV [9]. Viral RNA replication and protein expression follow as the viral proteins assemble and bud from the cell aided by viral neuraminidase (NA). Importantly, during IBV replication, the viral non-structural 1 (NS1) protein is involved in counteracting immune recognition by RIG-I, which is a cytosolic pattern recognition receptor (PRR) responsible for the type-1 interferon (IFN) response [9], and NS1 inhibits IFNβ [9].

It is unclear why IBV is mostly a human pathogen. The limited instances in which IBV infects other animals likely represent reverse zoonosis events. It is known that IBV can interact with human and murine but not avian homologs of host proteins that support viral replication, and this feature possibly contributes to the mammalian cell tropism of IBV [10,11]. Both IBV lineages have undergone antigenic drift by accumulating escape mutations in the head domains of the HA and NA proteins. Intriguingly, the B/Yamagata lineage viruses have predominantly acquired NA mutations, while B/Victoria lineage viruses have predominantly undergone HA changes, including numerous deletions in known antigenic sites [12], which has forced the inclusion of these strains in recent influenza vaccines. Despite the use of a quadrivalent vaccine that includes one IBV strain from each B/Yamagata and B/Victoria lineage, vaccine effectiveness is low, e.g., ~50% [13]. As most IBV strains are avirulent in mouse models and only transiently produce low virus titers in the lungs [14], it has been difficult to immunologically assess IBV strains. This is the reason why this study examined the immune response to non-mouse-adapted B/Florida/04/2006 (B/Yamagata lineage). Additionally, B/Florida/04/2006 was previously included in the Fluzone Quadrivalent vaccine, which contained A/California/07/2009 (H1N1), A/Victoria/210/2009 (H3N2), B/Brisbane/60/2008 (Victoria lineage), and B/Florida/04/2006 (Yamagata lineage). 

To help determine the features of immunity and disease pathogenesis linked to B/Florida/04/2006 infection in a B6 mouse model, dexamethasone (DEX) was used to reduce the immune and inflammatory response so that it would be possible to determine whether morbidity and mortality were predominantly immune- or viral-mediated. DEX is a corticosteroid and is widely used to control inflammation. The anti-inflammatory effects of DEX are complex, but primarily, the mechanism of action is mediated by the inhibition of inflammatory cells and the suppression of the expression of inflammatory mediators [15]. Specifically, DEX treatment alters the cytokine and chemokine milieu (e.g., IL-8) through inhibition of nuclear factor kappa-light-chain-enhancer of activated B cells (NF-κB) [16]. DEX treatment alters T cell responses through a reduction in proliferation and differentiation, a mechanism likely mediated by a decrease in CD28 (a co-stimulatory molecule important for T cell function) [17]. 

A previous study showed that DEX treatment of mice infected with 10^5.5^ TCID_50_ of H1N1 IAV resulted in reduced inflammation and morbidity [18], suggesting effects attributed to immune-mediated disease. However, in another study, DEX-treated mice infected with H5N1 cleared IAV by day 8 post-infection, similar to the untreated group, and there was no effect on morbidity or mortality [19]. Interestingly, DEX-treated mice infected with H5N1 cleared the virus by day 8 pi, similar to the untreated group, suggesting that morbidity and mortality were likely immune-mediated, as there were no substantial differences in inflammatory cytokines or immune markers except for a small reduction in lymphocytes in the DEX-treated mice compared with untreated controls. Thus, differences in DEX efficacy attributed to concentration or treatment differences or IAV strain differences can affect the results.

In this study, B6 mice were treated daily with DEX, which, as expected, delayed viral clearance and dampened the immune and pro-inflammatory response to B/Florida/04/2006 infection. The results showed that, compared with untreated, IBV-infected B6 mice, the immune response was a mediator of morbidity, mortality, and disease pathogenesis and that lung viral clearance was likely associated with a CTL response, while the lung leukocyte influx likely contributed to the disease. DEX-treated B6 mice received daily intraperitoneal (i.p.) administration (10 mg/kg) of DEX starting the day before infection (D-1) and continuing through the end of the study (D8). Control mice received daily i.p. PBS administration. As anticipated, DEX treatment delayed viral clearance while reducing histopathology, but interestingly, it did not improve the overall survival of the mice. PBS control mice had significantly less virus in their lungs by day 8 pi and a more robust immune response associated with higher lung histopathology. Despite these differences, both groups had a considerable immune response, although the magnitude of the response was less robust in DEX-treated mice compared with PBS-treated mice. This response included a substantial and sustained influx of bronchoalveolar lavage (BAL) leukocytes, including macrophages, neutrophils, T cells, and Th1-type cells, as well as modified expression of proinflammatory genes by lung cells. PBS-treated mice also had considerably increased BAL leukocytes compared with DEX-treated mice, suggesting that lung pathology is mediated in part by increased leukocyte numbers—particularly CD8+ T cells. Taken together, these data suggest that B6 mice can model B/Florida/04/2006 infection, replication, and lung disease, which is principally immune-mediated and is accompanied by virus-mediated mortality. 

## 2. Materials and Methods

### 2.1. Virus and Infection

B/Florida/04/2006 was provided by Ted Ross from the University of Georgia. Master stocks of IBV were authenticated by RNA sequencing and neutralization assays to confirm the strain. The virus was passaged twice in Madin–Darby Canine Kidney (MDCK) cells (ATCC CCL-34). Briefly, MDCK cells were propagated to 80% confluency in DMEM (Gibco, Waltham, MA, USA) + 10% FBS (Hyclone, Logan, UT, USA). The cells were subsequently washed twice with PBS (Gibco), then replenished with DMEM (Gibco) containing penicillin–streptomycin (Sigma Aldrich, Burlington, MA, USA) and 4% Bovine Serum Albumin (Sigma Aldrich). The cells were inoculated with B/Florida/04/2006 in the presence of TPCK-Trypsin (ThermoFisher Scientific, Waltham, MA, USA). The cells were incubated for 3 days at 37 °C until CPE was observed. The supernatants were harvested and centrifuged at 4 °C to pellet the cellular debris. The supernatant was stored frozen at −80 °C. 

### 2.2. Mice and Tissue Collection

Male B6 mice (10–12 weeks old) were obtained from Jackson Laboratories (Bar Harbor, ME, USA). The mice were given water and food ad libitum. The mice were treated with either dexamethasone (DEX) (10 mg/kg) or PBS intraperitoneally. The DEX was diluted in PBS. Mice were treated daily starting 24 h prior to infection and continuing until the end of the study. 

The mice were anesthetized using avertin (2,2,2-tribromoethanol; Sigma) and i.n. infected with 10^3^ PFU/mouse B/Florida/4/2006 in PBS, which was determined to be the lowest lethal inoculum. A clinical sign scoring system was used that included lethargy (1), dyspnea (2), body weight loss <15–20% of original weight (1), and body weight loss ≥ 20% of original body weight (3). Mice that accumulated a clinical score of 3 were euthanized. Four mice from each group were sacrificed on days 1, 2, 4, 6, and 8 pi, and their lungs were harvested. The lungs were transferred to gentleMACS tubes (Miltenyi Biotech, Westphalia, Germany) containing DMEM (Gibco) supplemented with penicillin–streptomycin (Gibco). The gentleMACS™ is a benchtop mechanical dissociator that creates viable single-cell suspensions or homogenizations. Briefly, the lungs were homogenized in 2 mL of DMEM (without added enzymes) using a GentleMACS Dissociator using the preset program for mouse lungs (program: Mouse Lung 02_01, 38 s mechanical dissociation) (Miltenyi Biotec) followed by centrifugation at 400× *g* for 10 min at 4 °C. The supernatant was frozen and stored at −80 °C. Organs were collected on day 8 pi and included the heart, kidney, liver, sera, spleen, or brain, and each organ was mechanically dissociated using the gentleMACS pre-programmed, mouse organ-specific program, centrifuged at 400× *g* for 10 min at 4 °C, and plaqued in the same fashion as the lung homogenate. All homogenates were stored at −80 °C. 

All infected mice were weighed daily. Any mouse that exceeded 20% weight loss of its original weight or displayed severe clinical symptoms was euthanized. Mice were also scored on the basis of clinical signs and euthanized if their clinical score reached 3 or higher. All procedures were performed in accordance with the UGA institutional animal care and use committee (IACUC approval 03-006-Y2-A0).

### 2.3. Determination of Lung Viral Titers

Organ homogenates (i.e., lung, heart, kidney, liver, sera, spleen, or brain) were thawed at RT, and then 10-fold serial dilutions of the homogenates were overlaid on MDCK cells. The MDCK cells were at 95–100% confluency at the time of assay. Organ homogenate samples were incubated for 60 min at RT with gentle mixing every 15 min. After 60 min, the serial dilutions were removed, and the MDCK cells were washed with DMEM. The wash medium was removed and replaced with a mixture of plaque media and 1.6% agarose (Sigma). Plaque media contained MEM, HEPES, L-Glutamine, and penicillin–streptomycin (all from Gibco). The MDCK cells were incubated at 37 °C with 5% CO_2_ for 48 h. After 48 h, the agarose overlay was removed, and the cells were washed with PBS. MDCK cells were fixed with 10% neutral buffered formalin for a minimum of 15 min. The formalin was discarded, and the MDCK cells were stained using 1% crystal violet (Sigma). The MDCK cells were then washed with distilled water to remove the crystal violet. Plaques were then counted, and the PFU/mL titer was calculated using the number of colonies and the dilution factor. The limit of detection for viral plaque titers was 50 PFU/mL. 

### 2.4. Bronchoalveolar Lavage (BAL)

BAL was collected from the trachea by inserting a catheter (Exel Int., Redondo Beach, CA, USA) and flushing the lungs with 1 mL ice-cold PBS three times. Once collected, BAL was centrifuged at 500× *g* × 10 min at 4 °C to separate leukocytes and BAL fluid (BALF), which was frozen at −80 °C until cytokine and chemokine analysis. Cells were resuspended in FACS buffer (0.8% BSA/PBS) and counted using a hemocytometer and Trypan blue exclusion method [20]. 

### 2.5. Flow Cytometry

BAL cells were washed in FACS buffer and resuspended in Fc block (anti-CD16/CD32, Pharmingen, San Diego, CA, USA) on ice. Cells were stained with appropriate concentrations of anti-CD3-Alexa Fluor647, anti-CD11b-Alexa Fluor488, anti-CD11c-PE, anti-DX5-PerCP/Cy5.5, anti-Ly6G-PE/Cy7, or isotype controls (all from BD Biosciences, Franklin Lakes, NJ, USA, BioLegend, San Diego, CA, USA, or eBiosceince, San Diego, CA, USA). BAL cells were washed in FACS buffer, fixed with 2% PFA (Ted Pella, Redding, CA, USA), and washed 1× with FACS buffer. For CTL examination, BAL cells were washed and resuspended in stimulation media (RPMI-1640 media containing 10% FBS, 1X antibiotics/antimycotics (Gibco), 25 ng/mL phorbol 12-myristate 13-acetate (PMA) (Sigma Aldrich), 1.25 μg/mL ionomycin (Sigma Aldrich), and 10 ug/mL BFA (BD Biosciences)) to stimulate the production of and retain cytokines and incubated at 37 °C for 4 h. Following stimulation, cells were washed 1× in FACS buffer, blocked in Fc block, and stained with anti-CD3-PerCP/Cy5.5, anti-CD4-APC, and anti-CD8-PE or isotype controls (BD Biosciences, BioLegend, or eBiosceince). BAL cells were washed and fixed with 1% PFA, permeabilized with 1X BD Perm/Wash solution (BD Biosciences), and stained with anti-IFNγ-Alexa Fluor488 (BD Bioscience) and anti-granzyme B-PE/Cy7 (eBioscience) or isotype controls. Cells were washed in Perm/Wash solution (BD Biosciences) and resuspended in FACS buffer for analysis. Fixed cells were stored at 4 °C for no longer than 24 h. Leukocyte phenotypes expressing CD3^−^, CD11b^+^, and Ly6G^-^ were considered macrophages; neutrophils were considered CD3^−^, CD11b^+^, Ly6G^+^, T cells (CD3^+^), T helper cells (CD3^+^, CD4^+^, CD8^−^), and cytotoxic T lymphocytes (CD3^+^, CD4^−^, CD8^+^). The distribution of cell surface markers was determined using a BD LSRII (Becton Dickinson, Mountain View, CA, USA), and compensation was performed using Comp beads (Invitrogen) and analyzed using FlowJo (BD Biosciences) using >20,000 events.

### 2.6. PCR and Gene Expression

Whole mouse lungs were removed and stored in RNAlater (Sigma Aldrich) at 4 °C until analysis. To isolate RNA, organs were rinsed in sterile PBS to remove RNAlater followed by homogenization using gentleMACS tubes (Miltenyi Biotech, Westphalia, Germany) and 2 mL nuclease-free water (ThermoFisher). A portion of supernatant was used for RNA isolation using NEB Monarch RNA Cleanup Kit as described by the manufacturer (New England Biolabs, Ipswich, MA, USA). cDNA synthesis reactions were performed using LunaScript RT Supermix (New England Biolabs) as described by the manufacturer. To determine gene expression level changes in the target genes of interest (Table 1), qPCR reactions were performed using the 2X SYBR Green qPCR Master Mix low ROX Reference dye kit (Agilent, Santa Clara, CA, USA). ΔΔCT PCR analysis comparing cycle threshold values for the genes of interest and a reference gene (i.e., beta-actin (ACTB), which has been used successfully in previous lung cytokine gene expression experiments in mice infected with influenza virus [21,22,23,24]) allowed for relative expression fold change data to be analyzed, as previously described [25,26]. The Minimum Information for the Publication of Quantitative Real-Time PCR Experiments (MIQE) was followed as described by Bustin et al. [27], and qPCR was performed with ~100% efficiency as determined by serial dilution validation experiments (data not shown). 

### 2.7. Histopathology

Lungs were collected on days 6 and 8 pi for histopathology determination. Lungs were inflated with 10% neutral buffered formalin (NBF, Fisher Scientific, Hampton, NH, USA) through the trachea, sutured, and stored in NBF for at least 24 h. Fixed lungs were embedded in paraffin, sectioned at a 4.0 μm thickness, mounted on positively charged glass slides, and stained with hematoxylin and eosin (H&E), and coverslips were added. Histological sections were evaluated blindly by a board-certified veterinary pathologist. Briefly, peribronchiolitis (inflammatory cells surrounding bronchioles), perivasculitis (inflammatory cells (e.g., lymphocytes) surrounding small blood vessels), interstitial pneumonitis (increased thickness of alveolar walls often attributed to neutrophil influx), and alveolitis (primarily neutrophils and macrophages within the alveolar space) were reviewed [28,29,30]. 

### 2.8. Statistics

All statistical analyses are specified in figure legends and were performed using Prism 9 (GraphPad Software, San Diego, CA, USA). Data are expressed as the mean ± standard error of the mean (SEM). Experiments were repeated two independent times, and representative data are shown. Statistical analyses are specified in the figure legends. 

## 3. Results

### 3.1. DEX Treatment Does Not Reduce Mortality

The 50% mouse lethal dose (MLD_50_) was determined using the method of Reed and Muench [31]. Mice were anesthetized and i.n. infected with dilutions of B/Florida/04/2006 to determine the MLD_50_, which was established to be 3 × 10^2^ PFU (Figure 1). Mice infected with ≥10^3^ PFU developed severe weight loss that resulted in euthanasia by day 8 pi (Figure 1). To determine the immune and pro-inflammatory features attributed to IBV infection, mice were infected with B/Florida/04/2006 at the lowest lethal dose (10^3^ PFU), and one group was treated with DEX to deter viral clearance and reduce immunity and inflammation.

The lung viral titers were determined on days 1, 2, 4, 6, and 8 pi in PBS- and DEX-treated B6 mice (Figure 2A). Overall, there was a significant difference (*p* < 0.05) between the PBS- and DEX-treated mice in viral lung titers (Figure 2A). While DEX-treated mice still had high lung titers (>10^4^ PFU) on day 8 pi, PBS-treated mice had mostly cleared detectable lung virus loads by day 8 pi. Despite this difference, both groups of mice continued to lose body weight, reaching ≥20% body weight loss, which required euthanization (Figure 2B,C). This outcome indicates that the lung virus load does not necessarily correlate with mortality and suggests that the immune response was likely a contributor to disease. Recall that DEX reduces the immune and pulmonary inflammatory response [32,33], and treatment reduced lung viral clearance but did not prevent mortality, revealing a virus-mediated role in disease and mortality. Thus, it is likely that both immune- and virus-mediated diseases have roles in disease pathogenesis. 

### 3.2. BAL Cell Response to B/Florida/04/2006 Infection and DEX Treatment

To determine the BAL cell response to B/Florida/04/2006 infection, BAL was collected on days 1, 2, 4, 6, or 8 pi and analyzed (Figure 3). DEX- and PBS-treated mice had similar total numbers of BAL cells through day 4 pi, representing the peak of lung viral replication (Figure 2A), but beginning on day 6 pi through day 8 pi, PBS-treated mice had higher numbers of total BAL cells compared with DEX-treated mice. There were no statistically significant differences between PBS- and DEX-treated mice at any time point. The numbers of BAL cells were quantified for their expression of phenotype markers of leukocyte subpopulations by flow cytometry (Figure 4). Cells were gated by FSC/SSC, and the leukocytes were determined: macrophages were considered CD3^−^, CD11b^+^, Ly6G^−^, neutrophils (CD3^−^, CD11b^+^, and Ly6G^+^), T cells (CD3^+^), T helper cells (CD3^+^, CD4^+^, and CD8^−^) and cytotoxic T lymphocytes (CD3^+^, CD4^−^, and CD8^+^). 

In PBS-treated mice, T cells were significantly (*p* < 0.05) and consistently increased at all time points examined (Figure 4A). Notably, the percentage of T cells was significantly (*p* < 0.05) increased on days 4 and 6 pi following DEX treatment; however, as DEX treatment continued overtime, the number of T cells in these mice dropped with no statistically significant difference between DEX-treated, IBV-infected, and untreated, uninfected control mice by day 8 pi. CD11b^+^ macrophages were significantly (*p* < 0.05) increased in PBS- and DEX-treated mice on days 4, 6, and 8 pi compared with those in untreated, uninfected B6 mice (Figure 4B). DEX- and PBS-treated mice also had higher percentages of Ly6G+ neutrophils at all time points examined (Figure 4C). As expected from Figure 3, macrophages and neutrophils were increased substantially in PBS-treated compared with DEX-treated mice, indicating that DEX reduces this innate response to IBV infection. CD11c^+^ and DX5^+^ NK cell populations were not affected or increased during infection (data not shown). The increased numbers of pulmonary macrophages and neutrophils likely contribute to lung disease pathogenesis, as studies have implicated macrophages and neutrophils as contributors to IAV-mediated pathology through pro-inflammatory cytokine expression and/or reactive oxygen species (ROS) [34,35,36]. As T cell numbers were noted to be more treatment-dependent than macrophages and neutrophils, these cells were further investigated.

On day 8 pi, the CTL to Th ratio in the BAL was 2:1 (Figure 5A), a finding similar to that found in a recent study of IAV-infected mice [37], and there were significantly (*p* < 0.05) more CTLs in PBS-treated mice than in DEX-treated mice. To determine the BAL T cell functionality, BAL cells collected on day 8 pi were nonspecifically stimulated with PMA/ionomycin, and intracellular levels of granzyme B and IFNγ were determined. The BAL cells were analyzed by flow cytometry and lymphocyte-gated, i.e., SSC^lo^FSC^lo^/CD3^+^ and CD4^+^, CD8^+^, and ‘cytokine’^hi^. The results showed that 20% of PBS-treated and 10% of DEX-treated CTLs were positive for granzyme B (Figure 5B), while 20% of PBS-treated and 25% of DEX-treated CD8+ cells expressed IFNγ and 30% of CD4^+^ T cells produced IFNγ, regardless of treatment (Figure 5C). The appreciably fewer granzyme-B-expressing CTLs in the DEX-treated compared with PBS-treated mice was not unexpected, as DEX treatment has been shown to modify the immune response by preventing granzyme B transcription [38,39]. Interestingly, another paper describing DEX in the RSV mouse model showed IFN-independent downregulation of mucosal responses, suggesting that DEX, in this model, downregulated granzyme B and not IFNγ [40]. Taken together, these data suggest that IBV infection induces a robust BAL cell response in which DEX treatment reduces the CTL response. 

### 3.3. Lung Gene Expression in Response to IBV Infection

Understanding host gene expression specific to IBV infection can aid the understanding of immunity and disease pathogenesis. Specific gene expression patterns (or signatures) reflect the biological state and can serve as biomarkers. The expression of lung mRNA transcripts was determined by PCR from B6-infected mice early (day 2 pi) or later (day 8 pi) after infection to determine the relative gene expression changes using primer pairs noted in Table 1 [26]. Early gene expression (day 2 pi) in PBS-treated mice showed upregulated Th1-type (TBET and IFNG) genes, while DEX-treated mice expressed upregulated IFN (Figure 6A). TBET is a T-box gene encoding transcription factors involved in the regulation of developmental processes and is the human ortholog of the mouse Tbx21/Tbet gene [41]. Studies in mice showed that TBET is a Th1 cell-specific transcription factor that controls the expression of the hallmark Th1 cytokine, IFNγ [42]. By contrast, PBS-treated mice displayed upregulated granzyme B (GZMB) and perforin (PRF1) transcripts, which are CTL-associated responses that induce apoptosis of virally-infected cells [43]. On day 8 pi, PBS-treated B6 mice had a 12-fold increase in TBET and IFNG transcript expression, while DEX-treated mice had reduced (2–4-fold) expression of TBET but a 32-fold increase of IFNγ (IFNG) (Figure 6B). This large increase is consistent with a previous report showing that dexamethasone reduced inflammation without altering IFN responses during RSV infection in mice [40]. Lungs from PBS-treated mice had increases in GZMB and PRF1 transcript expression, which is consistent with the CTL functional findings reported here. DEX-treated mice had similar increases. There was a 250-fold increase in GZMB and a 16-fold increase in PRF1 transcript expression in the PBS-treated mice, as well as a 125-fold and a 32-fold increase in GZMB and PRF1 transcript expression in the DEX-treated group, respectively. Interestingly, TNFA (TNFα), a canonical pro-inflammatory cytokine, was not substantially upregulated on day 2 pi regardless of treatment as previously observed [44], and DEX treatment downregulated TNFA transcript expression. By contrast, on day 8 pi, there was considerable upregulation of TNFA in the PBS-treated group. Taken together, the transcript expression analysis suggests that CTL- and Th1-like genes are substantially upregulated on day 8 pi. 

### 3.4. DEX Treatment Reduces Lung Histopathology Associated with IBV Infection

To determine if IBV morbidity and mortality in B6 mice were linked to lung histopathology, the lungs of IBV-infected mice were collected and examined on day 8 pi (Figure 7). PBS-treated, IBV-infected mice developed severe lung disease characterized by bronchiolitis, alveolitis, vasculitis, and inflammation, while DEX-treated mice had less severe histopathology. This finding is consistent with the reduced inflammatory BAL cells in DEX-treated mice (Figure 3 and Figure 4) and, in particular, the reduced CTLs compared with PBS-treated mice (Figure 5A,B). These results suggest that in the absence of DEX, mice succumb primarily to immune-mediated disease (e.g., CTL), as these mice presented with pronounced histopathological lesions despite virus clearance. However, DEX treatment dampens the immune response, precluding IBV clearance and preventing an over-reactive immune response, leading to mortality. 

## 4. Discussion

Influenza viruses continuously emerge as drift variants, which can cause annual epidemics and periodic pandemics; however, little is known about IBV compared to IAV. Research has demonstrated the need for universal immunity to IAV and IBV strains [45]; however, this can only be achieved by a better understanding of the immunity and disease pathogenesis of these viruses. Despite the substantial disease burden in humans associated with IBV infection, studies in animal models are limited, as IBV infection is less permissive than IAV [46]. In this study, the LD_50_ for B/Florida/04/2006 in B6 mice was determined and shown to be 3 × 10^2^ PFU/mouse, which caused disease (e.g., weight loss and lethargy) with 100% mortality occurring at 10^3^ PFU/mouse or higher inoculums. Most infected mice progressively lost weight until day 6 pi and thereafter exhibited dramatic weight loss that resulted in some death or the need for euthanasia (≥20% weight loss). 

As predicted, an inoculum exceeding 10^3^ PFU/mouse resulted in rapid weight loss, which corresponded to high lung viral loads and the mice succumbing to infection between days 4 and 6 pi. Interestingly, lung IBV titers of mice inoculated with 10^3^ PFU were below the limit of detection by viral plaque assay on day 8 pi, yet IBV infection was lethal; therefore, we hypothesized that disease and mortality were mediated in part by an aberrant immune response. To address this possibility, mice were i.p. administered 10 mg/kg DEX daily starting on day −1 before infection. DEX treatment has been shown to reduce the immune and pro-inflammatory responses [47]; thus, we expected to reduce immune-mediated disease. As expected, DEX inhibited lung virus clearance, but DEX-treated mice also lost weight and died at the same time as the PBS-treated mice. IAV mouse studies have shown both virus- and mouse strain-dependent effects associated with DEX treatment. For example, DEX prevented H1N1 IAV-mediated lung damage in B6 mice, as determined by lung histology following H1N1 infection, suggesting that DEX is protective [18]. One caveat in comparing this study to ours is that the investigators saw weight loss as high as ~30%, whereas our IACUC protocol required humane euthanasia in mice at 20% body weight loss. Because mice were euthanized at 20% body weight loss in our study, we cannot note DEX as ameliorating mortality. We speculate that. if we had no humane euthanasia requirement, despite the lack of robust viral clearance, mice may have recovered from severe weight loss without major damage to the lungs. In another study, DEX did not ameliorate disease in H5N1-infected BALB/c mice [19]. Similar to our results, both PBS- and DEX-treated mice had increased pulmonary lymphocytes and neutrophils, with slightly fewer cells in DEX-treated mice. Interestingly, there was a notable decline in macrophages, which may be attributed to differences between the virus (i.e., IAV H5N1 vs IBV) and/or the method of macrophage quantification, as the above report utilized cell smears, while macrophages in our study were identified by extracellular markers using flow cytometry. 

Evaluation of BAL cell infiltrates hinted that macrophages and neutrophils may contribute to the pathology through the direct action of cytokines [48], the release of reactive oxygen species and/or NETosis [49], and/or through the recruitment and activation of T cells. BAL T cells recruited to the lungs were of similar composition to those in previous IAV studies [37]; however, the T cell response was appreciably lowered in the DEX-treated mice compared with that in PBS-treated mice. Moreover, there was a significant (*p* < 0.01) reduction in CTL functionality concerning the CD8+ CTL to secrete granzyme B in DEX-treated mice. Th1-like responses were similar between DEX and PBS treatments. Importantly, lung gene transcript expression was consistent with the cellular phenotype and functional data, with T cell responses including TBET, IFNG, GZMB, and PRF1 upregulation throughout infection. 

The lungs from PBS-treated IBV-infected mice were markedly diseased compared with the lungs from DEX-treated, IBV-infected mice. PBS-treated mice had a more robust immune response, particularly by CTLs, which potentially contributed to lung virus clearance and histopathological damage. By contrast, DEX treatment reduced lung damage, likely mediated in part by a reduced inflammatory response; however, DEX treatment did not prevent weight loss (disease) and mortality. It is likely that DEX-treated mice succumbed to virus-mediated disease, while PBS-treated mice succumbed to immune-mediated disease. To investigate the possibility of extrapulmonary viral dissemination, the heart, spleen, brain, sera, kidney, and liver were collected, and the levels of replicating virus were determined by plaque assay. No replicating virus was detectable in any of the organs examined, indicating that there was likely no systemic IBV infection in DEX-treated mice that could have contributed to disease (data not shown); rather, IBV was localized in the lung and was likely the key mediator of disease in DEX-treated mice. Two important limitations of this study were 1) non-lung organs were not plaqued prior to 8 dpi, and thus we cannot exclude transient viremia and/or extrapulmonary dissemination and 2) IHC was not performed on lung or non-lung organs to determine whether viral replication was occurring elsewhere, e.g., in pneumocytes. Future studies should investigate potential histopathological abnormalities in non-lung organs and the IHC of these organs to detect IBV replication. 

These findings show that B/Florida/04/2006 can infect B6 mice, providing a model to allow the development of a better understanding of IBV immunity and disease pathogenesis and showing evidence for robust inflammatory and immune responses to IBV that can lead to morbidity and mortality. 

## Figures and Tables

**Figure 1 vaccines-10-01440-f001:**
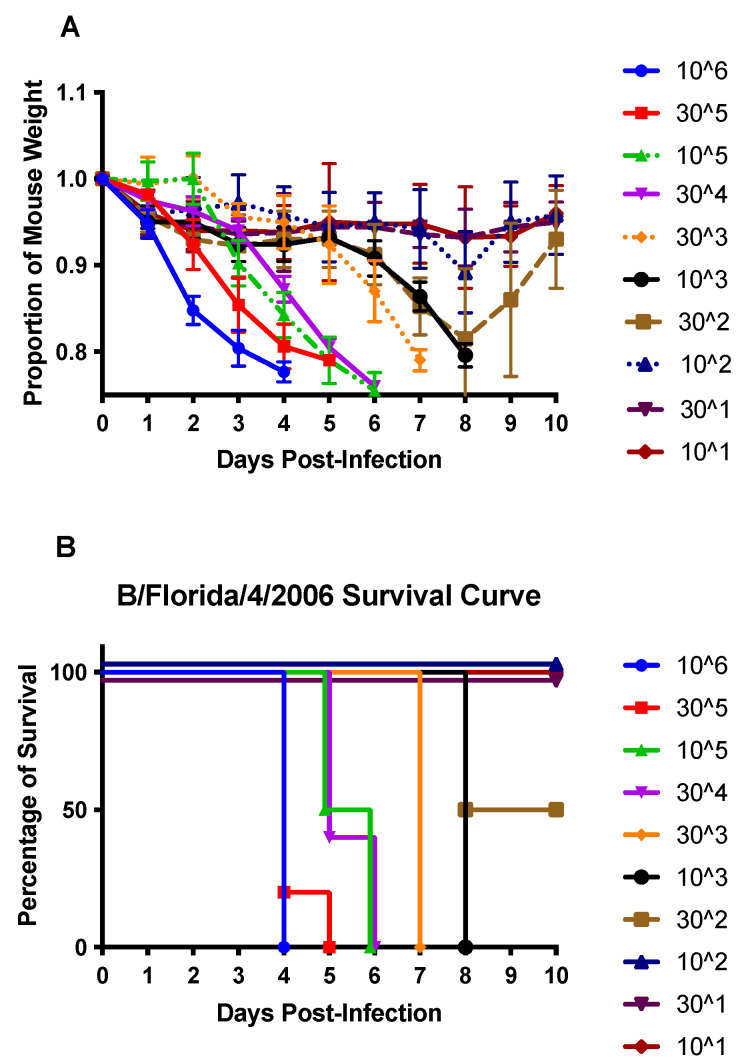
Weight loss (**A**) and survival (**B**) of B/Florida/04/2006-challenged C57BL/6 (B6) mice. B6 mice were i.n. infected with 10^6^ or half-log dilutions down to 10^1^ PFU of B/Florida/04/2006. Weight loss and survival were recorded daily for ten days post-infection. Points represent the mean +/− SEM of the proportion of original weight (n = 5 mice/group).

**Figure 2 vaccines-10-01440-f002:**
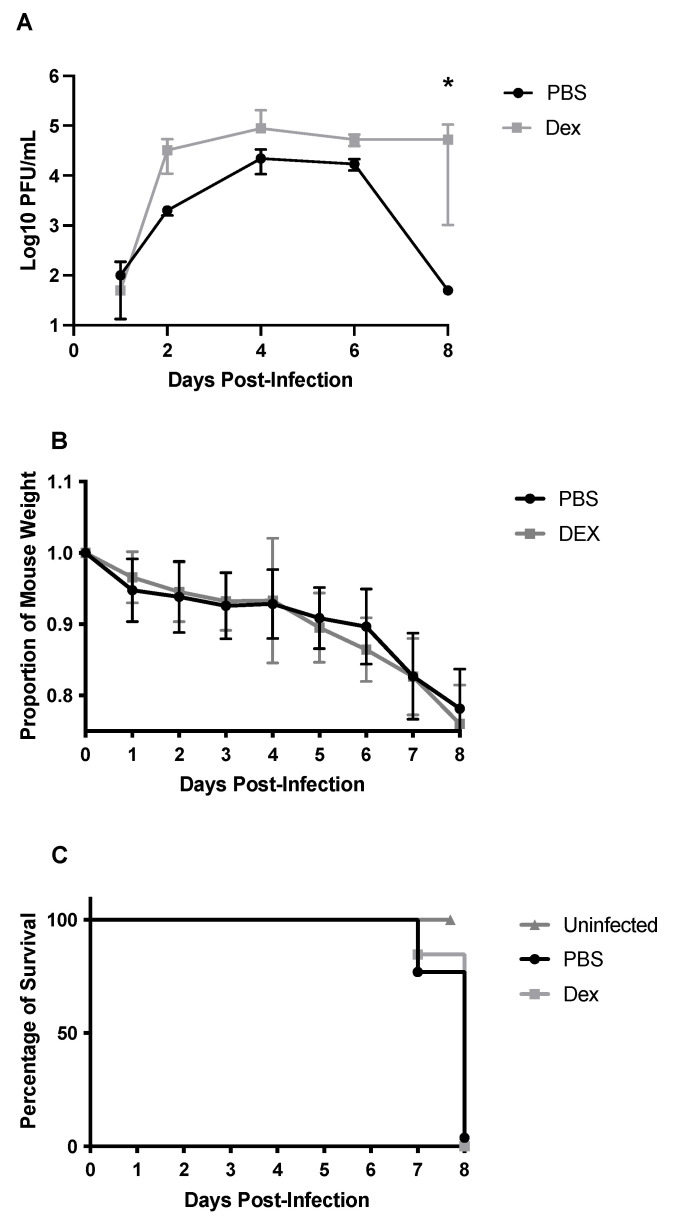
Viral lung titers (**A**), weight loss (**B**), and survival (**C**) of B/Florida/04/2006-challenged B6 mice. B6 mice were i.n. challenged with 10^3^ PFU B/Florida/04/2006. Viral lung titers were determined on days 1, 2, 4, 6, or 8 pi by virus plaque assay. Dots represent the mean PFU/mL of lung homogenate +/− SEM. The limit of detection for viral plaque assay was 50 PFU/mL. Weight loss and survival were recorded daily for 8 days post-infection. Points represent mean +/− SEM of the proportion of original weight. * *p* < 0.05 between the PBS and DEX-treated mice by Multiple Mann-Whitney U tests (n = 5 mice/group).

**Figure 3 vaccines-10-01440-f003:**
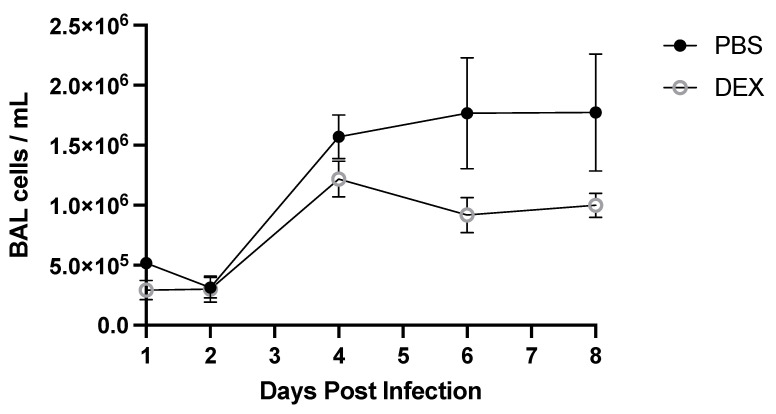
Total BAL cells during IBV infection. Mice were treated with PBS or 10 mg/kg DEX daily (starting on day −1 before infection), and BAL cells were collected at indicated days pi with IBV B/Florida/04/2006. Counts were performed using a hemocytometer and Trypan blue exclusion. Data represent the mean + SEM of live BAL cells/mL (n = 3–5 mice/group/time point). No significance between PBS and DEX-treated mice on any day post-infection was detected by multiple Mann–Whitney U Tests.

**Figure 4 vaccines-10-01440-f004:**
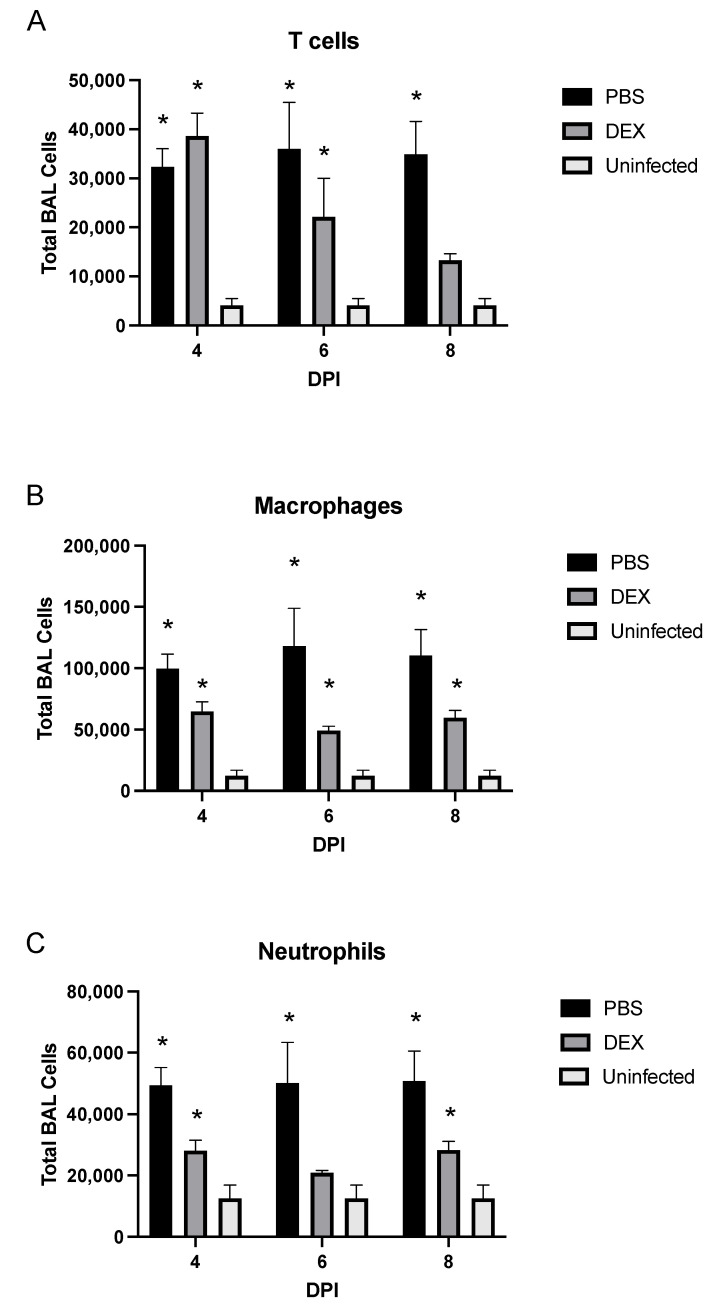
BAL cell phenotypes during IBV infection. BAL cell phenotypes from mice treated with PBS (black bars) or 10 mg/kg DEX (gray bars) and uninfected, untreated mice (white bars) were stained for (**A**) CD3^+^ T cells, (**B**) CD3^−^/CD11b^+^/Ly6G^−^ macrophages, and (**C**) CD3^−^/CD11b^−^/Ly6G^+^ neutrophils, as determined by flow cytometry. Bars represent the mean +/− SEM of total BAL leukocytes on days 4, 6, and 8 pi (DPI). * *p* < 0.05 by two-way analysis of variance (ANOVA) with Bonferroni’s correction compared with uninfected, untreated mice. (n = 3–5 mice/group/time point).

**Figure 5 vaccines-10-01440-f005:**
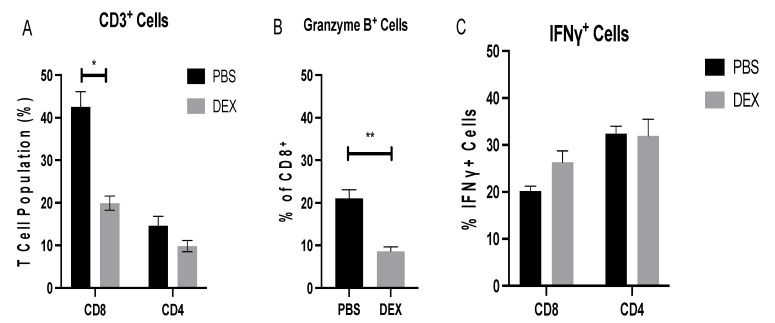
BAL T cell response during IBV infection. BAL T cells from mice treated with PBS or 10 mg/kg DEX and infected with 10^3^ PFU IBV were collected on day 8 pi and analyzed by flow cytometry. (**A**) Extracellular surface staining to determine percentages of CD8^+^ and CD4^+^ cells of CD3^+^ gated lymphocytes. A portion of BAL cells was stimulated with PMA/Ionomycin as described in materials and methods, and functional analyses of cytokine production were determined for (**B**) granzyme B and (**C**) IFNγ. Bars represent the mean +/− SEM, * *p* < 0.05, ** *p* < 0.01 by multiple Mann–Whitney U test.

**Figure 6 vaccines-10-01440-f006:**
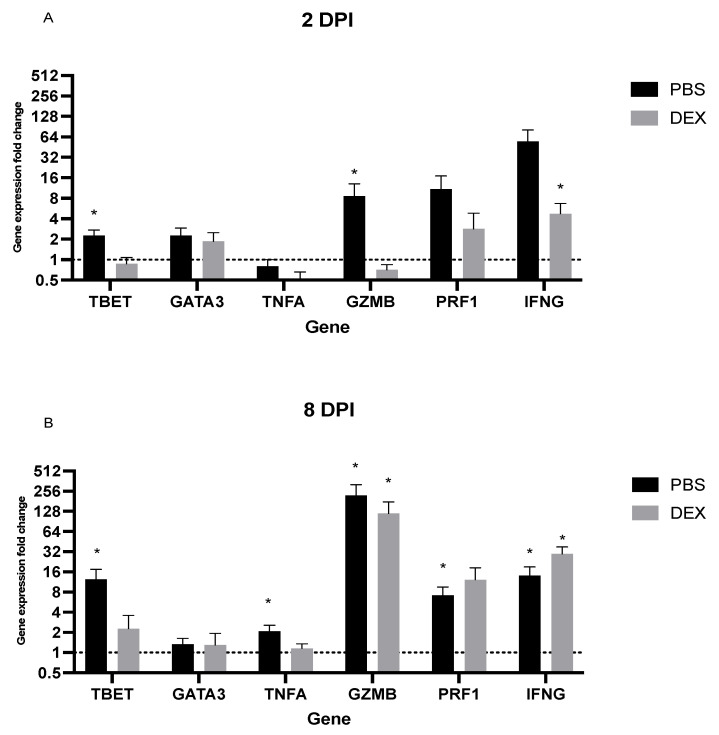
PCR lung transcript expression. Mouse lung transcript expression was performed from lungs of mice treated with PBS or 10 mg/kg DEX and infected with 10^3^ PFU IBV were collected (**A**) 2 and (**B**) 8 days post-infection (dpi). Bars represent the mean +/− SEM fold change of transcripts determined by ΔΔCt, normalized to ACTB, and uninfected, untreated controls, where * *p* < 0.05 by multiple Mann–Whitney U Tests.

**Figure 7 vaccines-10-01440-f007:**
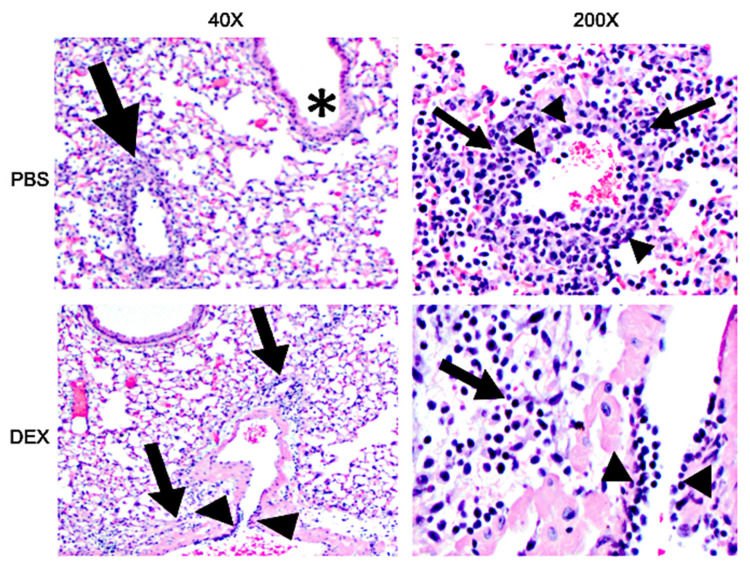
H&E staining of lungs: 40× (**left**) and 200× (**right**) magnification photomicrograph of a lung of a mouse that was treated with PBS (top) or DEX (bottom) and infected with IBV on day 8 pi. Bronchiolitis (asterisks), perivascular inflammation (arrows), vasculitis, (arrowheads).

**Table 1 vaccines-10-01440-t001:** Oligonucleotides used for gene expression experiments.

Gene	FWD	REV
TBX21	5′-TCC GGG AGA ACT TTG AGT CC-′	5′-TGG AAG GTC GGG GTA GAA AC-3′
GATA3	5′-GTG GTC ACA CTC GGA TTC CT-3′	5′-GCA AAA AGG AGG GTT TAG GG-3′
TNFA *		
GZMB *		
INFG	5′-ATG AAC GCT ACA CAC TGC ATC-3′	5′-CCA TCC TTT TGC CAG TTC CTC-3′
PFRN *		
ACTB	5′-AAG TGT GAC GTT GAC ATC CG-3′	5′-GAT CCA CAT CTG GAA GG-3′
IBV (NP)	5′-GGT TGG ACT TGA CCC TTC ATT A-3′	5′-CCA CTA AAG TTC CAC CTC CTT-3′

* Integrated DNA Technologies (Coralville, IA, USA) proprietary predesigned PrimeTime^®^ qPCR Primers.

## Data Availability

Data are available upon reasonable request.

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
