# Peer review of "Influenza B Virus (IBV) Immune-Mediated Disease in C57BL/6 Mice"

_vaccines, 2022, doi:10.3390/vaccines10091440_

Round 1

Reviewer 1 Report

In this manuscript, Bergeron et al. looked for immune-mediated versus virus-mediated outcomes in mice after influenza B virus infection and treatment with Dexamethasone. This study is well-performed. I have few minor comments:

1. In figure 1, it is difficult to interpret the colors. I recommend authors to change coloring scheme and for symbols also use 10^1 etc. 

2. In figure 4, make only one figure comparing different cell types between PBS and DEX.

Author Response

In this manuscript, Bergeron et al. looked for immune-mediated versus virus-mediated outcomes in mice after influenza B virus infection and treatment with Dexamethasone. This study is well-performed. I have few minor comments:

1. In figure 1, it is difficult to interpret the colors. I recommend authors to change coloring scheme and for symbols also use 10^1 etc. 

Thank you, we have updated the figure

2. In figure 4, make only one figure comparing different cell types between PBS and DEX.

Thank you, we have made Figure 4 into one figure.

Reviewer 2 Report

Whereas influenza B viruses (IBV) that cause seasonal epidemics in humans do not have an animal reservoir, unexpectedly C57BL/6 mice infected intranasally with a low multiplicity of infection of B/Florida/04/2006 IBV developed substantial morbidity and mortality.  In their search to understanding the mechanisms of disease, particularly involving immune-mediated versus virus-mediated outcomes, the authors demonstrate that dexamethasone (DEX)-treated mice had a lower pro-inflammatory response and reduced lung pathology despite the presence of high viral lung titers.  Additionally, mortality was comparable to vehicle-treated mice indicating that mortality is linked to lung virus replication and associated cytopathology in the absence of substantial immune and pro-inflammatory response (i.e. cytotoxic CD8+ T lymphocytes).  Extrapulmonary viral dissemination (serum, heart, liver, spleen, kidney and brain at day 8 post-infection) was not detected by viral plaque assay. Therefore, it appears that DEX-treated mice succumbed to virus-mediated lung disease, while vehicle-treated mice succumbed to immune-treated lung disease (bronchiolitis, vasculitis and perivascular inflammation).

Although this work suggests that the mouse model of B/Florida/04/2006 IBV infection may help advance our understanding of IBV immunity and pathogenesis, the methods and results sections of this manuscript often lack essential information necessary to understand the experimental design and validate the results and conclusions.  Also, additional experiments are needed to definitely exclude the possibility of extrapulmonary viral dissemination and organ damage.

Line 18-25: switch the order of the sentences. Present the results first (line 21-25) before the conclusions (line 19-21).  Define all acronyms at their first mention in the abstract and main text.

Line 115: how were the B/Florida/04/2006 IBV stocks characterized?

Line 128: were mice infected intranasally with dilutions of unpurified B/Florida/04/2006 IBV stocks in cell culture medium?  Since control mice were treated with phosphate-buffered saline (PBS), have you considered the effects of the cell culture medium dilutions on the mouse immune responses separately from those of B/Florida/04/2006 IBV?

Line 129: describe the treatment of mice with DEX (dose, vehicle, route of administration, timing relative to viral infection). Some of that information is in the legend of Figure 3.

Line 136: describe the lung homogenization process using gentleMACS (mechanical or enzymatic) and the centrifugation of the homogenates (relative centrifugal force, duration)? 

Line 138-139: did you titer B/Florida/04/2006 IBV in serum, heart, liver, spleen, kidney and brain before day 8 post-infection to rule out viremia?  Did you perform histological examination at different times after infection to rule out whether organ damage contributed to morbidity and mortality?

Line 161: when was bronchoalveolar lavage performed after intranasal infection and/or DEX treatment?

Line 169-170, 179-180, 182-183: clearly indicate to which antibody each fluorophore is conjugated.

Line 212-214: elaborate on the scoring of the histological findings including reference(s).

Line 216: describe in sufficient detail which statistical analyses were performed. Some statistical methods are mentioned in the legends of Figure 4 and 5.

Table 1: provide a more complete, descriptive title. Cite Table 1 in the text.

Figure 1A: label with “Proportion of mouse weight”.  Figure 1B: revise the labeling of the mouse survival plots to clearly identify the different doses of B/Florida/04/2006 IBV.

Figure 2A: label with “Proportion of mouse weight”.

Figure 4A, 4B: were the fold change relative to values at the time of infection or of DEX administration?

Figure 7: provide clear histological images. The images are not in focus.

Author Response

Line 18-25: switch the order of the sentences. Present the results first (line 21-25) before the conclusions (line 19-21).  Define all acronyms at their first mention in the abstract and main text.

Thank you, this has been addressed in the revised manuscript.

Line 115: how were the B/Florida/04/2006 IBV stocks characterized?

We are not certain what is meant by ‘characterized’. B/Florida/04/2006 was provided by Ted Ross from the University of Georgia as indicated in the manuscript. The virus was sequenced and propagated as described, and then was plaqued multiple times to confirm titer.  The virus has also been used in neutralization assays.  We would be happy to answer any additional questions about the B/Florida/04/2006 virus.

Line 128: were mice infected intranasally with dilutions of unpurified B/Florida/04/2006 IBV stocks in cell culture medium?  Since control mice were treated with phosphate-buffered saline (PBS), have you considered the effects of the cell culture medium dilutions on the mouse immune responses separately from those of B/Florida/04/2006 IBV?

The control mice were treated with PBS (as opposed to DEX) which is unrelated to the inoculum media.  While the virus was grown in DMEM, the virus was diluted multiple logs in sterile PBS to obtain a 103 PFU titer.  DMEM is less than 1% of the inoculum used to infect the mice.  We do not expect this extremely low amount of DMEM (less than 0.5uL) to have any substantial effect on the mouse or immune system.  

Line 129: describe the treatment of mice with DEX (dose, vehicle, route of administration, timing relative to viral infection). Some of that information is in the legend of Figure 3.

We appreciate the reviewer’s observations - the text has been updated in the revised manuscript.

Line 136: describe the lung homogenization process using gentleMACS (mechanical or enzymatic) and the centrifugation of the homogenates (relative centrifugal force, duration)? 

We have added a description to the revised manuscript. The gentleMACS™ are benchtop mechanical dissociators that create viable single-cell suspensions or homogenizations Over 500 publications are available by the manufacturer using the gentleMAC in virus research showing the devices can assist in many ways. Briefly, the lungs were homogenized in 2 mL of DMEM using a GentleMACS Dissociator using the preset program for lungs (Miltenyi Biotec) followed by centrifugation at 400xg for 10 minutes. The supernatant was frozen and stored at −80°C

Line 138-139: did you titer B/Florida/04/2006 IBV in serum, heart, liver, spleen, kidney and brain before day 8 post-infection to rule out viremia?  Did you perform histological examination at different times after infection to rule out whether organ damage contributed to morbidity and mortality?

We did not titer non-lung organs prior to day 8. We would that expect if extrapulmonary dissemination occurred the virus would be detectable at 8 dpi in DEX-treated mice as high lung viral titers were noted at this time. We cannot rule out earlier dissemination nor microscopic histopathological damage to organs however no gross histopathology was detected.

Line 161: when was bronchoalveolar lavage performed after intranasal infection and/or DEX treatment?

 All timepoints are relative to infection, and DEX treatment has been clarified in the methods.

Line 169-170, 179-180, 182-183: clearly indicate to which antibody each fluorophore is conjugated.

This has been added to the revised manuscript.

Line 212-214: elaborate on the scoring of the histological findings including reference(s).

Thank you, references to previous histological findings are included.

Line 216: describe in sufficient detail which statistical analyses were performed. Some statistical methods are mentioned in the legends of Figure 4 and 5.

We note that all statistics are now included in legends.

Table 1: provide a more complete, descriptive title. Cite Table 1 in the text.

We have revised the title in the revised manuscript and cite the Table in the text.  

Figure 1A: label with “Proportion of mouse weight”.  Figure 1B: revise the labeling of the mouse survival plots to clearly identify the different doses of B/Florida/04/2006 IBV.

Thank you, we have updated the figure.

Figure 2A: label with “Proportion of mouse weight”.

Thank you, we have updated the figure

Figure 4A, 4B: were the fold change relative to values at the time of infection or of DEX administration?

These values were relative to mock animals sacrificed on the same day post-infection as indicated.

Figure 7: provide clear histological images. The images are not in focus.

The images were collected in focus. Their sharpness has been modified to enhance the image.

Round 2

Reviewer 2 Report

Line 19-22, 318-320: Figure 7 DEX appears to show reduced immune cell response, but not an “absence of substantial immune and pro-inflammatory response” following dexamethasone.  Also, what is the histological evidence of the “lung virus replication and associated pathology damage”?  In other words, does influenza B/Florida/04/2006 replicate in and damage the lung parenchyma (e.g., pneumocytes)?  Detection of the virus in lung tissue sections by immunohistochemistry would help better understand the mechanism of disease and death.

Line 34-35, 39-40: clarify these related statements: ”IBV infection averages a quarter of the annual influenza disease burden” and “both strains [IAV and IBV] contribute equally to hospitalization burden and complications.”

Line 77-79, 283-284: describe in more detail the relative effects of dexamethasone on different types of immune cells and inflammatory mediators.

Line 115: mention that your B/Florida/04/2006 stocks were authenticated by RNA sequencing and neutralization assays.

Line 123-124: what was the range of your B/Florida/04/2006 titer stocks? How many fold were your stocks diluted prior to administration to mice?  The concern is not the quantity of DMEM, but the amount of MDCK canine cell lysate and bovine serum albumin in the diluted virus stock cell lysate that may be immunogenic in mice but is duplicated in the PBS-treated mice.

Line 139-144: no need to advertise for the GentleMACS manufacturer. However, since this instrument is commonly used to obtain viable single-cell suspensions, what protocol did you use to "homogenize" the different mouse organs (enzyme mix H, R and A; duration, etc.)?

Line 169-170: describe the centrifugation method used to separate BAL leukocytes.

Line 184, 187: specify BAL cells.

Line 199-200: elaborate on how the mouse lungs were preserved in RNAlater and homogenized. Were the lungs inflated or perfused with RNAlater prior to harvesting? In what medium were the lungs homogenized using GentleMACS to isolate total RNA?

Line 220: add a brief description of the scoring scale to avoid the need to refer to other references.

Line 257, 258, 272: Figure 4 does not have Panel A and B.

Line 319-320: “DEX-treated mice had minimal to no histopathology”.  This description is not supported by Figure 7 DEX, in which “perivascular inflammation (arrows) and vasculitis (arrowheads)” are highlighted.

Line 341: given the experimental design (3 experimental groups, multiple time points), Student’s T-test is inappropriate. If the data have a parametric distribution, analysis of variance followed by post-hoc tests (e.g., Tukey) must be used.

Line 351, 366, 372: in Figure 3, 5 and 6, given the experimental design (multiple groups and time points), Mann-Whitney U test is inappropriate. If the data have a parametric distribution, Kruskal-Wallis H test must be used.

Line 410-413, 79-84: compare in quantitative terms your experimental results with those reported in references 16 and 17.

Line 432-437: since you have not examined non-lung organs prior to dpi 8, it would be important to state that you cannot rule out earlier virus dissemination and histopathological damage to these organs.  An example of such early effects has been reported: Wu XX, et al. The viral distribution and pathological characteristics of BALB/c mice infected with highly pathogenic influenza H7N9 virus. Virology Journal 2021;18:237.

Round 3

Reviewer 2 Report

Further review of your manuscript has identified significant issues related to experimental methodology and statistical analysis.

Line 207-210: you report that whole mouse lungs were stored in RNAlater and subsequently homogenized in nuclease-free water using gentleMACS instrument to isolate total RNA.  However, according to the RNAlater manufacturer, tissue must be cut to a maximum thickness of 0.5 cm in at least one dimension in order to allow RNAlater to quickly permeate the tissue and inactivate RNAses. Also, according to the gentleMACS manufacturer’s protocol “Homogenization of tissue for total RNA isolation”, the tissue should be homogenized quickly in a “lysis/binding buffer” (e.g. Trizol, Qiazol, RLT); nowhere does it say water.  As a result, lung total RNA samples were undoubtedly badly degraded by endogenous RNases present inside areas of whole lungs not permeated by RNAlater and by intracellular RNAses unleashed during mechanical homogenization of lung tissue in the absence of RNase inhibitors. Therefore, all results (Fig. 6) and conclusions related to PCR analysis of lung gene expression cannot be relied upon and must be removed from the manuscript.

Line 232: you still did not explain the scale 0 to 3. To what percentages of inflamed lung area corresponds 0, 1, 2 and 3? For example, score 0, 0%; score 1, 1-20%; score 2, 21-50%; score 3, 51-100%.

Line 233-237: summarize the statistical analyses performed for each and every experiments in the Statistics Section. 

Fig. 2, 3, 4 and 6: as an example, let’s consider the results shown in Fig. 3. At line 261-264, you wrote “DEX- and PBS-treated mice had similar total numbers of BAL cells through day 4 pi representing the peak of lung viral replication (Figure 2A), but beginning on day 6 pi, PBS-treated mice had higher numbers of total BAL cells compared to DEX-treated mice”.  Note that you compare the number of BAL cells among different time points in order to determine the peak and subsequent time-associated changes in BAL cell numbers. In your reply letter you explain that “T-test was used to compare individual days between only the DEX and PBS treatment groups” and “we only compared the PBS and DEX at each time point… Each time point was independently analyzed…” It is always incorrect to perform the same statistical test repeatedly at different time points because 1 test out of 20 will yield a false positive result (p<0.05) plus the fact that you compare results among different time points based upon multiple independent tests.  I believe the appropriate test in your case is two-way ANOVA (measured variable x time points) for time series. Since the samples were collected on different mice at different time points, repeated measure ANOVA cannot be used.

Author Response

Line 207-210: you report that whole mouse lungs were stored in RNAlater and subsequently homogenized in nuclease-free water using gentleMACS instrument to isolate total RNA.  However, according to the RNAlater manufacturer, tissue must be cut to a maximum thickness of 0.5 cm in at least one dimension in order to allow RNAlater to quickly permeate the tissue and inactivate RNAses. Also, according to the gentleMACS manufacturer’s protocol “Homogenization of tissue for total RNA isolation”, the tissue should be homogenized quickly in a “lysis/binding buffer” (e.g. Trizol, Qiazol, RLT); nowhere does it say water.  As a result, lung total RNA samples were undoubtedly badly degraded by endogenous RNases present inside areas of whole lungs not permeated by RNAlater and by intracellular RNAses unleashed during mechanical homogenization of lung tissue in the absence of RNase inhibitors. Therefore, all results (Fig. 6) and conclusions related to PCR analysis of lung gene expression cannot be relied upon and must be removed from the manuscript.

We do not agree with the reviewer’s statements. We do not wish to debate these statements and therefore have removed figure 6 and its conclusions from the manuscript as requested.

Line 232: you still did not explain the scale 0 to 3. To what percentages of inflamed lung area corresponds 0, 1, 2 and 3? For example, score 0, 0%; score 1, 1-20%; score 2, 21-50%; score 3, 51-100%.

We have removed the mention of scoring in the materials and methods because we do not describe scores in the text.

Line 233-237: summarize the statistical analyses performed for each and every experiments in the Statistics Section.

We have included statistical analysis for each experiment in the statistics section and clarified this in the legends.

Fig. 2, 3, 4 and 6: as an example, let’s consider the results shown in Fig. 3. At line 261-264, you wrote “DEX- and PBS-treated mice had similar total numbers of BAL cells through day 4 pi representing the peak of lung viral replication (Figure 2A), but beginning on day 6 pi, PBS-treated mice had higher numbers of total BAL cells compared to DEX-treated mice”.  Note that you compare the number of BAL cells among different time points in order to determine the peak and subsequent time-associated changes in BAL cell numbers. In your reply letter you explain that “T-test was used to compare individual days between only the DEX and PBS treatment groups” and “we only compared the PBS and DEX at each time point… Each time point was independently analyzed…” It is always incorrect to perform the same statistical test repeatedly at different time points because 1 test out of 20 will yield a false positive result (p<0.05) plus the fact that you compare results among different time points based upon multiple independent tests.  I believe the appropriate test in your case is two-way ANOVA (measured variable x time points) for time series. Since the samples were collected on different mice at different time points, repeated measure ANOVA cannot be used.

As requested, we have adjusted our statistical analysis for figures 2, 3, and 4 to two-way ANOVA. Figure 6 is now deleted.